

# Pressure-induced transitions in FePS$_3$: Structural, magnetic and electronic properties

Shiyu Deng[1*], Siyu Chen[1], Bartomeu Monserrat[1,2],
Emilio Artacho[1,3,4†] and Siddharth S. Saxena[1,5‡]

**1** Cavendish Laboratory, University of Cambridge, J. J. Thomson Avenue,
Cambridge, CB3 0HE, United Kingdom
**2** Department of Materials Science and Metallurgy, University of Cambridge,
27 Charles Babbage Road, Cambridge CB3 0FS, United Kingdom
**3** CIC Nanogune BRTA and DIPC, Tolosa Hiribidea 76, 20018 San Sebastian, Spain
**4** Ikerbasque, Basque Foundation for Science, 48011 Bilbao, Spain
**5** British Management University, 22a Sebzar Street, Tashkent, Uzbekistan

⋆ sd864@cam.ac.uk , † e.artacho@nanogune.eu ,
‡ sss21@cam.ac.uk

## Abstract

FePS$_3$ is a prototype van der Waals layered antiferromagnet and a Mott insulator under ambient conditions, which has been recently reported to go through a pressure-induced dimensionality crossover and an insulator-to-metal transition. These transitions also lead to the appearance of a novel magnetic metallic state. To further understand these emergent structural and physical properties, we have performed a first-principles study using van der Waals and Hubbard $U$ corrected density functional theory including a random structure search. Our computational study attempts to interpret the experimental coexistence of the low- and intermediate-pressure phases and we predict a novel high-pressure phase with distinctive dimensionality and different possible origins of metallicity.



# 1   Introduction

Two-dimensional (2D) materials with tunable electrical and magnetic properties have substantial potential in the design of atomically thin devices for data storage, quantum computing [1] and clean energy generation related with photocatalytic water splitting applications [2]. For several decades the study of 2D phenomena in magnets has been inhibited due to particular interpretations of the well-known Hohenberg-Mermin-Wagner theorem [3, 4]. However, this is beginning to change given the recent observation of intrinsic long-range magnetic order in 2D materials, including the ferromagnetic (FM) conductor $Fe_3GeTe_2$ [5], magnetic insulator $Cr_2Ge_2Te_6$ [6] and antiferromagnetic (AFM) insulators $TM\mathrm{PX}_3$ ($TM$ = Mn, Fe, Ni, V, etc., and $X$ = S, Se) [7–9] which are the focus of this study. It has been found that magnetic anisotropy could open up an excitation energy gap to counteract the enhanced thermal fluctuations in low dimensional materials [10]. Different from conventional thin films [11], 2D materials bound only by van der Waals (vdW) interactions offer more precise manufacturing control and reproducibility [12, 13].

A recently realized advantage of 2D magnets is that they are often materials where the ground state is dominated by the physics of strong correlations. These materials thus become fertile ground for exploring novel phases and emergent phenomena. For instance, the well-known high-$T_c$ cuprates are obtained via doping the 2D AFM precursors [14,15]. Most recently, the pressure-induced superconductivity (SC) in the vicinity of AFM order has attracted much interest, with prominent examples being the iron-based layered compound LaFeAsO [16], FeSe [17,18], and other transition-metal compounds e.g. MnSe [19], CrAs [20] and $AuTe_2Br$ [21]. In such systems, reduced dimensionality is believed to enhance SC [22, 23]. However, the understanding of the fundamental physics of these emergent phenomena is still hindering proper theoretical formulation. In this regard, the $TM\mathrm{PX}_3$ family represents a promising new avenue of research.

Transition metal phosphorous trichalcogenides $TM\mathrm{PX}_3$ have proven to be ideal examples of exploration through tuning of the structural, magnetic and electronic properties in 2D vdW layered systems exhibiting both long-range magnetic order and strong correlations [24, 25]. Being Mott or charge-transfer insulators at ambient pressure, the band gap of $TM\mathrm{PX}_3$ could be tuned systematically via the application of pressure. Recent high-pressure studies have revealed spin-crossover transitions, insulator-to-metal transitions (IMT) and even the emergence of SC in these compounds [26–31]. In this family, $FePS_3$ is an ideal prototype to start with, considering the Ising nature of the magnetic moments of Fe ions [32, 33], small band gap of about 1.5 eV, lowest resistivity of $1.0 \times 10^{12}\,\Omega\,\mathrm{cm}$ [34] at ambient pressure and experimental evidence for the evolution of the structural, electronic and magnetic properties at high pressures [28–31].

For FePS$_3$, it has been challenging to accurately ascertain its crystallographic and electronic properties due to the lack of precise atomic positions from the experimental data. More generally, correlation effects always represent a challenge for computational simulations. While experimentally observed spin-spin short-range correlations are hard to tackle with first-principles calculations, it would be worthwhile to carefully examine these effects, particularly near the transition from 2D vdW compounds to more 3D bonded configurations.

In 2018, Haines *et al.* [28] and Wang *et al.* [29] performed independent high pressure experiments on FePS$_3$ powder samples. Both groups observed the IMT and volume collapse in response to the external pressure but proposed incompatible models for the high-pressure (HP) phase. Wang *et al.* [29] claimed that the low-pressure (LP) monoclinic symmetry remains until the HP region and the in-plane lattice collapse contributed the most to the volume collapse during the iso-structural phase transition at $\sim 13$ GPa. They also reported that when the HP phase turns into a metallic phase in the case of FePSe$_3$, which is a related compound with similar structural and magnetic properties, SC was observed at 2.5 K and 9.0 GPa [29]. Meanwhile, Haines *et al.* [28] claimed that there are two transitions. The first happens around 4 GPa via inter-planar sliding. The LP phase evolves into HP-I without symmetry and dimensionality change and remains insulating. The next occurs around 14 GPa with an interlayer lattice collapse and the bulk symmetry changed to $P\bar{3}1m$ in HP-II. The HP-II phase was determined to be metallic and more 3D-like. Subsequently, two computational studies by Zheng *et al.* [35] and Evarestov *et al.* [36] did not come up with consistent conclusions regarding the origins of IMT or the impact of magnetic configurations on the crystal structure. A later Raman spectroscopy work by Das *et al.* has detected two phase transitions at 4.6 GPa and 12.0 GPa, complementing the previous experimental observations and models [31]. The most recent study by Jarvis *et al.* [37] compares the different experiments utilizing powder samples with and without the helium pressure medium, and includes the single crystal diffraction results in the discussions. The experimental environment plays an essential role in the high-pressure behaviour.

The detailed magnetic structure of FePS$_3$ is challenging to resolve even at ambient pressure [32, 38–40] and affects the high-pressure phases as well. A recent study by Coak *et al.* [41] further characterised the LP, HP-I and HP-II phases of FePS$_3$ up to about 18 GPa. They first examined the evolution of magnetic phases with pressure using powder neutron diffraction, and proposed that from LP to HP-I the interlayer interaction transforms from antiferromagnetic to ferromagnetic. In the metallic HP-II phase, the long-range magnetic order is suppressed while a form of short-range order emerges [41]. This is a rather different result from the spin crossover transition model proposed by Wang *et al.* [29], where the final state is claimed to be non-polarised. The magnetic interactions in $TMPX_3$ are complex and offer further opportunities to explore the underlying physics [42–44].

To understand how external pressure tunes the dimensionality, structural, electronic and magnetic properties in FePS$_3$, it is essential to grasp the systematic evolution of the crystalline phase with applied pressure. Given the fact that different experimental environments and setups affect the high-pressure behaviour, we re-examine the high-pressure structures via a random structure search method using density functional theory. The advantage of this method is that it does not require empirical knowledge of the experimental findings and thus allows us to search for the most stable and metastable structures from the energy considerations. We are able to reproduce the proposed phases and predict a novel one in the high-pressure region. Moreover, we look into the interlayer sliding and dimensionality change in FePS$_3$ as it undergoes the phase transitions in detail, together with the evolution of electronic and magnetic properties.

## 2  Methods

First principles calculations are performed based on density functional theory (DFT), using the Cambridge Serial Total Energy Package (CASTEP) code [45, 46], version 19.1. The generalized-gradient-approximation (GGA) within the framework of Perdew, Burke, and Ernzerhof (PBE) [47] is used for the exchange-correlation functional. We utilize the "on-the-fly" [48] generated ultrasoft pseudopotentials based on the formalism of Vanderbilt [49], as implemented in CASTEP. The error achieved by this set is 0.4 meV/atom within the test framework of Lejaeghere *et al* [50].

The Broyden-Fletcher-Goldfarb-Shanno (BFGS) algorithm [51] was used for the geometry optimization, with the force convergence tolerance being 0.05 eV/Å. The vdW interactions have been taken into account using the Tkatchenko-Scheffler approach [52]. Correlation effects have also been considered within the framework of the DFT+$U$ method [53–55], allowing for spin polarization. Different starting spin configurations of the self-consistency (SCF) cycle allow for different spin polarization results and the local spin arrangements among Fe atoms. Here, both the SCF and structural relaxations are constrained in the relative spin orientations, yet without constraint on the magnitude of local spins, allowing for magnitude variations on prescribed spin arrangements (using CASTEP's SPIN_FIX and GEOM_SPIN_FIX option for SCF and relaxations) that are later compared in energy. For the localized $d$ orbitals of the Fe atoms a Hubbard effective electron-electron repulsion $U$ is taken to be 2.5 eV, as used in similar studies [35, 36, 56]. The effect of varying this parameter in the results is included in the results and discussion below.

The convergence criterion for the overall formation energy has been set as $10^{-6}$ eV/atom for all enthalpy calculations. To ensure proper convergence within reasonable computational time, we perform a series of tests running the self-consistent single-point simulations. Tests are shown in Appendix A. Valence-electron wave-functions are described with a plane-wave basis set, with a cutoff energy for the expansion of 550 eV, as selected from the tests in Appendix A. The sampling of discretized $k$-points across the Brillouin zone follows the Monkhorst-Pack scheme [57] and is determined to be 0.03 Å$^{-1}$ along each axis (CASTEP's KPOINTS_MP_SPACING). The Fourier transform grid for the electron density is larger than that of the wave functions by a factor of 2.0.

To search for the stable and metastable structures of FePS$_3$ at high pressures, we perform an *ab initio* random structure search (AIRSS) [58] to generate random structures that are fully relaxed with CASTEP [45, 46]. We carry out the AIRSS structure search at 0, 10 and 20 GPa using randomly generated structures containing between 1 and 4 chemical formula units of FePS$_3$ within one simulation cell. A random set of unit cell lengths and angles is generated as the cell volume is re-normalized to a random value within ±50% of the volume derived from the known structures. Atomic positions are also selected at random within the cell, with the constraint of the minimum distance being 3.3 Å for Fe-Fe, Fe-P and S-S, and of 2.4 Å, 2.1 Å and 2.0 Å for Fe-S, P-P, and P-S, respectively. A full BFGS relaxation is then performed by imposing hydrostatic pressure (by introducing a target stress tensor for the desired hydrostatic pressure in CASTEP) arriving at a relaxed structure and corresponding energy, and enthalpy. About 1500 initial structures were generated across the potential energy surface (PES) at each pressure point. The method cannot ensure that the global minimum is found as the search cannot be exhaustive within a finite computation time, as is well known for global minima searches [58–61]. Yet, the adequacy of our search is supported by us finding the expected stable phases several times, the so-called 'marker' structures. In our case, they are the known low-pressure (LP) phase, and the first and second high-pressure phases (HP-I and HP-II), as characterised in the literature [28].

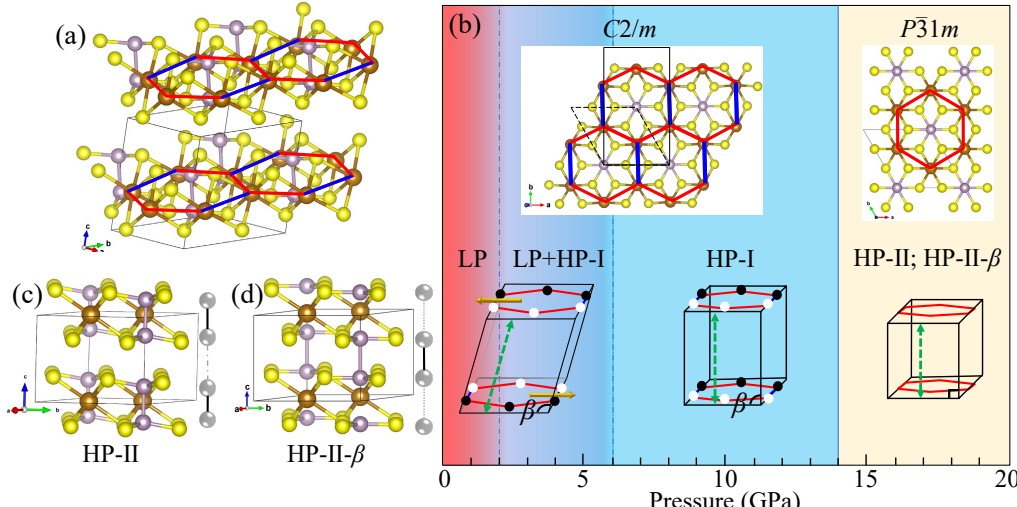

Figure 1: Crystal structure of various FePS$_3$ phases and phase diagram vs pressure. (a), (c) and (d) show the LP, HP-II and HP-II-$\beta$ structures. (b) Pressure phase diagram Ref. [41] and random structure search results in this paper. The red and blue rods indicate the ferromagnetic (FM) and antiferromagnetic (AFM) coupling between the nearest neighbouring Fe ions, respectively. The black and white dots represent the magnetic moment on Fe$^{2+}$ pointing up or down, normal to the $ab$ plane. The LP and HP-I both host zigzag FM chains along $a$ axis and the neighbouring chains within each plane are AFM coupled. From LP to HP-I, the interlayer AFM coupling changes into FM instead. Below when we refer to them as (LP, AFM) and (HP-I, AFM) in contrast with the non-spin-polarized phases. The transition pressure value is sensitive to the magnitude of Hubbard $U$, for example it is $\sim 7.4$ GPa for $U = 2.50$ eV (see Fig. 2), and $\sim 11$ GPa for $U = 3.75$ eV figure (see Appendix B).

## 3 Results and Discussion

With the AIRSS search we manage to reproduce the known phases and in addition predict a new one in the high-pressure region. The novel structure shares the same template of the honeycomb and space group symmetry as HP-II, though it distinguishes itself with the characteristic P$_2$ bonding. The P atoms form stronger chemical bonds between the neighbouring layers instead of the intralayer configuration of previously proposed structures. There are hints of a similar evolution in the related material NiPS$_3$ [62]. The now fully connected 3D structure therefore differs from the weakly bound two-dimensional layers found in the HP-II phase. We refer to this phase as HP-II-$\beta$ going forward. Metastable phases with diverse building blocks are also predicted but in this work we focus on the phases in close proximity to the minimal enthalpy line, which happen to be close to experimental findings except for the mentioned HP-II-$\beta$ phase. Fig.1 summarizes the LP, HP-II, HP-I and HP-II-$\beta$ phases and a detailed discussion of the evolution with pressure is given in the subsequent section.

### 3.1 Pressure-induced Crystalline Phase Transitions

To establish the context in which the crystalline structure evolves with pressure, we start with the LP phase. FePS$_3$ has been subjected to a full X-ray structural characterisation since 1973, being the first among the family [63]. The neighbouring layers stack together and the bulk crystallized in the monoclinic space group $C2/m$ with an ideal $\beta$ angle, compared with the Mn and Ni counterparts [64], which suggest the least distortion within each layer. At ambient

pressure, FePS$_3$ is a Mott insulator with a band gap of $\sim 1.5$ eV. Across the family, the complicated competition among exchanges and anisotropy leads to different types of antiferromagnetic order with different directions for the collinear axes of the spin moments (Ising versus Heisenberg magnetic Hamiltonian). FePS$_3$ at ambient pressure has Ising-type AFM order with zigzag ferromagnetic (FM) chains along the $a$ axis coupled antiferromagnetically along the $b$ axis. The moments are normal to the $ab$ planes [63]. Being a 2D AFM with correlated physics, FePS$_3$ has been studied extensively in recent years with pressure being the tuning parameter.

Fig. 1 (a) displays the crystalline structure of LP FePS$_3$ with the Fe$^{2+}$ honeycomb being illustrated by a rigid rod. In the inset of Fig. 1 (b), the obtained spin up and down moments are indicated via black and white dots, respectively, defining ferromagnetically (FM) coupled lines of Fe atoms (red rods), which are antiferromagnetically ordered in respect to each other (blue rods). This is a striking result for a Fe essentially honeycomb arrangement, which being bipartite would not be expected to induce the kind of frustrations which give rise to the observed arrangement. It is, however, a known and expected result from the LP phase related to frustrations due to first vs longer range effective exchange couplings [39].

The magnetic configuration breaks the $C_3$ rotational symmetry and leads to slightly elongated Fe hexagons in the LP phase. The blue rods corresponding to the AFM coupling are 3.448 Å in length, while the red ones representing FM coupling are 3.426 Å. The intersite exchange within and in-between the planes is mediated through the surrounding P$_2$S$_6$ clusters, with P atoms centered within the distorted Fe hexagons. The shear in the interlayer coupling can also be considered to be behind the C3 symmetry breaking, but it is observed (see below) the symmetry breaking remains when the shear is removed by pressure. Defining the formula unit (f.u.) as [FePS$_3$], the primitive cell contains 2 f.u. in the cell, as indicated by the dotted line in the upper-left inset of Fig. 1 (b). The conventional cell consistent with previous literature involves 4 f.u. of [FePS$_3$], shown in solid black lines. For consistency with previous works, we adopt the conventional description of the lattice parameters with $a = 5.947$ Å, $b = 10.300$ Å, $c = 6.722$ Å and $\beta = 107.2°$ for the LP. The symmetry breaking distorts the Fe hexagons, showing sides along the zigzag spin chains that are shorter than the ones perpendicular to the chain, by a ratio of 0.65 % at ambient pressure. The $b/a$ ratio is, however, only very slightly affected, staying very close to the ideal $b/a = \sqrt{3}$.

With increasing pressure, the crystalline structure of LP evolves into new phases but some features of the LP are retained. The HP-I phase shares the same crystalline building block as LP, but differs in the stacking angle $\beta$ as neighbouring layers in LP slide towards each other. The $\beta$ angle turns nearly 90° in HP-I. In addition, though the in-plane magnetic configuration has been preserved, the interlayer coupling has changed from AFM to FM [41]. Both phases crystallise in the space group of $C2/m$. Experimentally a coexistence region of both LP and HP-I is observed from 2 to 6 GPa, from the powder diffraction data without pressure medium [28, 41]. Nevertheless, no coexistence was seen in the single crystal, or in the powder data with He pressure medium [37]. We discuss the transition from LP to HP-I in Section 3.1.1.

At the higher pressure region, the long-range magnetic order is suppressed and the Fe$^{2+}$ cations form perfect hexagons within the plane, as indicated in the upper-right inset of Fig. 1 (b). The recovery of the $C_3$ rotational symmetry at the centre of the Fe-hexagons leads to the symmetry crossover transition from $C2/m$ to $P\bar{3}1m$. The previously proposed HP-II and the predicted HP-II-$\beta$ share the same intralayer S-Fe-S template. The side-view for these two phases is shown in Fig. 1 (c) and (d), respectively. The P$_2$ bonding behaviour alters the dimensionality in HP-II from that in HP-II-$\beta$. It offers an ideal platform to explore the effect of dimensionality on the electronic and magnetic properties, which will be discussed in detail in Section 3.2.

In order to understand how these phases evolve with pressure, we performed first-principles geometry optimisation at various pressure points. The enthalpy-pressure phase di-

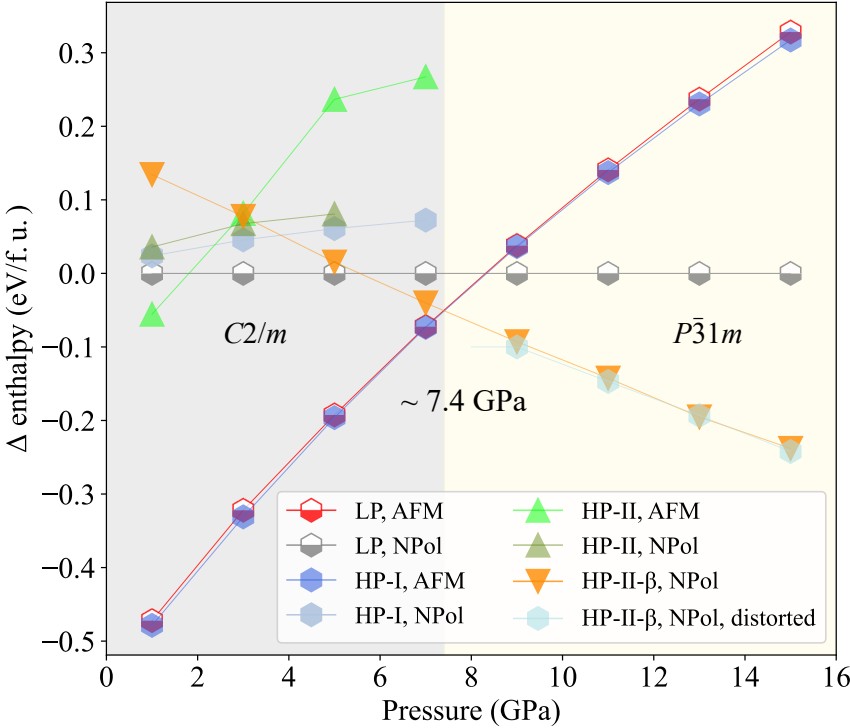

Figure 2: Pressure dependence of the enthalpy difference $\Delta H = H - H_{\text{ref}}$ of various phases with respect to the reference non-spin-polarized low-pressure phase (LP, NPol, half-grey hexagons). $\Delta H$ is shown for: the low-pressure phase with antiferromagnetically coupled spin chains (LP, AFM, half-red hexagons), high-pressure phase I, both AFM (blue hexagons) and non-polarised (HP-I, NPol, full light blue hexagons) and, high-pressure phase II, HP-II, both in the Néel AFM state (light green triangles) and NPol (dark green triangles) and the non-polarised HP-II-$\beta$ phase (orange triangles). The figure shows that below a transition pressure of $\sim 7.4$ GPa the antiferromagnetic LP and HP-I phases are very close in enthalpy, LP being very slightly favored at low pressures, but very close to unshearing to the HP-I phase (see Fig. 3). Above the transition pressure the HP-II phase becomes the stable one in its $\beta$ form, that is, with interlayer P-P bonds. The structure was relaxed allowing for a distortion breaking the triangular symmetry. The relaxation recovered the symmetry up to a residual distortion of 0.006 Å in the difference of inequivalent Fe-Fe nearest neighbour distances.

agram is summarised in Fig. 2. The enthalpy for the non-polarised LP phase has been chosen as the reference line. The spin polarisation is shown to be essential in establishing the relative stability of the various phases, with a very significant energy scale of up to half an eV/f.u. stabilizing the LP and HP-I phases. However, given their full polarization, both phases compete very closely in enthalpy, with differences LP and HP-I now on the meV/f.u. scale.

### 3.1.1 Coexistence of the LP and HP-I

The transition from LP to HP-I occurs via the relative sliding of the neighbouring layers and the change of interlayer magnetic coupling from AFM to FM. We display the enthalpy evolution in response to the interlayer sliding in LP and HP-I as a function of the $\beta$ angle in Fig. 3 at 1 GPa of pressure. The inset in the figure illustrates the shear in terms of the lattice viewed from the $b$ axis. For each phase, we manually slide the neighbouring layers and then relax the geometry with a fixed $\beta$ angle. The enthalpy of the fully optimised LP phase at $\beta = 107.8\,^\circ$

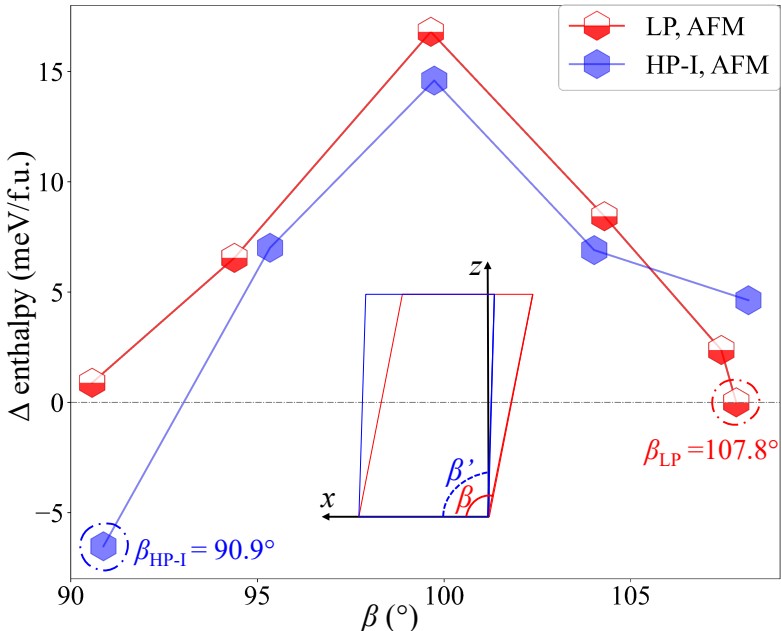

Figure 3: Enthalpy difference as a function of the $\beta$ angle for the LP and HP-I AFM phases of FePS$_3$ at 1.0 GPa. The enthalpy of LP relaxed at a fixed $\beta_{LP} = 107.8°$ is set as reference. The difference between both curves at the same pressure and shear is the different interlayer spin arrangement of the LP and HP-I phases.

is set as the reference. The first conclusion extracted from this plot is that already at this pressure the almost unsheared phase would be the stable one. Howeuver, a significant energy barrier ($\sim 15$ meV per f.u.) becomes very apparent in the figure for both phases, which could explain the experimentally observed coexistence of LP and HP-I shown in Fig. 1 (b) between 2 and 6 GPa, due to kinetic effects in the relaxation of the loading. In addition, experimentally the pressure cannot be ideally hydrostatic and phase coexistence might persist over a pressure range, depending on experimental conditions as it seems to have happened in experiments on powder samples [28].

### 3.1.2  The Dimensionality Change in HP-II and HP-II-$\beta$

Back in Fig. 2, there is a symmetry transition at $P \sim 7.4$ GPa from $C2/m$ (LP, HP-I) to $P\bar{3}1m$, which is qualitatively similar to the previous experimental findings. Such symmetry change also occurs when the bulk is reduced to monolayer thickness [65]. The trigonal symmetry for every single layer at that pressure is recovered by releasing the constraint of the monoclinic $\beta$ angle. When starting from a distorted geometry (HP-I), the structure relaxes towards the trigonal phase. The average in-equivalent Fe-Fe nearest neighbour distance is about 3.350 Å and the difference is 0.006 Å for the residual distortion. The enthalpy for the HP-II-$\beta$ phase with residual distortion is also shown in Fig. 2.

For the two trigonal phases, HP-II and HP-II-$\beta$ distinguish from one another in the P-P intra- and interlayer bonding behaviours. In order to investigate whether they are degenerate in energy or relax into the same local minimum after geometry optimisation, we explore the evolution of crystal structure including lattice parameters at the applied pressure, especially following the interatomic distance of P-P within and in-between the layers.

Fig. 4 shows the evolution of P-P distances within the layer (blue circles) and between the layers (green) for the relaxed HP-II phase from 1 to 15 GPa in steps of 2 GPa. The prediction is that HP-II transforms into the HP-II-$\beta$ phase around 9 GPa. The novel HP-II-$\beta$ phase is more

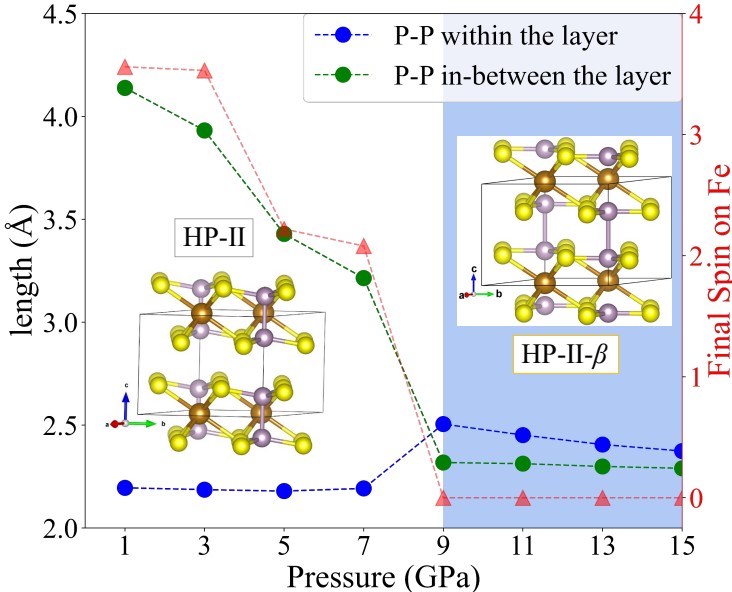

Figure 4: Evolution with the pressure of P-P interatomic distances for P nearest neighbors along the direction perpendicular to the layers, in the $P\overline{3}1m$ FePS$_3$ phase. Intralayer (interlayer) distance is shown as blue (green) circles. Red triangles show the localised spin on a Fe atom.

3D-like, compared to the HP-II phase. As far as we understand, it is difficult to experimentally determine the detailed P positions. In addition, further experimental evidence may be impeded by the substantial kinetic energy barrier expected for the change in connectivity, which implies P-P chemical bond restructuring.

Fig. 4 was obtained with the PBE functional, corrected by the vdW Trachenko-Scheffler approach, and a value of the Hubbard $U = 2.5$ eV on the Fe $d$ orbitals including spin polarization for the Fe atoms. We extensively explore the sensitivity of such a transition to the simulation parameters and approximations mentioned in Appendix B. The key prediction is that the dimensionality changes from 2D to 3D under pressure. Yet, increased values of Hubbard $U$ postpone the transition to higher pressure, bringing it into closer agreement with experiments, and thereby suggesting that the correlation strength in the FePS$_3$ system is quite substantial and makes a difference.

### 3.1.3 The Dynamical Stability of HP-II and HP-II-$\beta$

We have also explored the dynamical stability for HP-II and HP-II-$\beta$ phases under the same pressure before the transition occurs, here 7.0 GPa. The phonon spectrum has been calculated using the finite displacement method [66] in conjunction with nondiagonal supercells [67]. Multiple commensurate supercells have been first constructed where atoms are perturbed from equilibrium positions And then SCF calculations have been performed to evaluate the force-constant matrix. The finite displacement method allows the use of any electronic structure theory to obtain the lattice dynamics of FePS$_3$, taking the vdW interaction and Fe $d$-orbital correlation into account. A $3 \times 3 \times 3$ q-point grid has been adopted to sample the dynamical matrix of HP-II and HP-II-$\beta$ phases.

The phonon dispersions for the two phases are shown in Fig. 5 along a selected high-symmetry path within the Brillouin zone. Though there is no imaginary phonon at the zone center ($\Gamma$ point) for HP-II, the imaginary part along $\Gamma$ to A, highlighted in the shaded region, suggests certain instability for the HP-II structure along the vertical direction, which means that

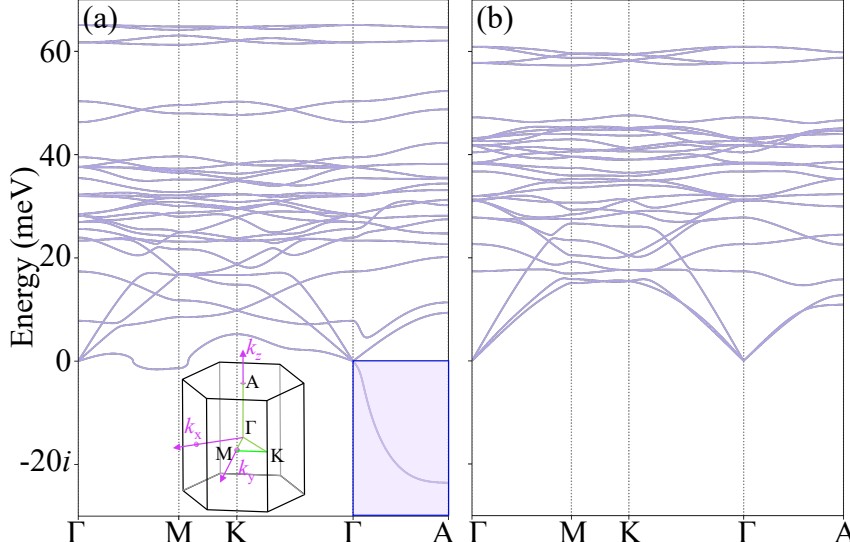

Figure 5: The phonon spectrum calculated within DFT+$U$ (Fe $d$: $U = 2.5$ eV) for fully relaxed (a) HP-II and (b) HP-II-$\beta$ phases at 7.0 GPa. The selected high symmetry path has been illustrated as an inset in (a).

the HP-II phase tends to distort itself along the direction to lower its energy. The symmetry constraint of the $P\bar{3}1m$ space group might give the reason why HP-II can still be found via the structure search and geometry relaxation before the transition occurs. By contrast, there is no imaginary phonon frequency across the whole Brillouin zone for the HP-II-$\beta$ phase, suggesting that this predicted new phase featuring P-P interlayer bonding geometry is indeed dynamically stable. This is consistent with the fact that a more compact structure along the vertical direction will be favored at high pressure. In addition, it is also worth noting that the volume of HP-II is about 8.8 % larger than that of HP-II-$\beta$, mainly arising from the $c$ lattice parameter being 5.407 Å for the former while being 4.915 Å for the latter, more 3D-connected phase. The P-P bonding length within the layer versus that in-between the layer is 2.192/3.215 for HP-II, and 2.605/2.310 for HP-II-$\beta$. The significant volume shrinkage along the vertical direction and the rearrangement of P-P bonding lead to the fact that the high-frequency optical modes in HP-II are softened in HP-II-$\beta$. We have observed that the optical modes around 63 meV refer to strong vibrations of P and S atoms with all Fe atoms remaining static. Those around 47 meV refer to the vibrations of predominantly S atoms. From the 2D layered structure (HP-II) to the 3D bonded structure (HP-II-$\beta$), the vibration modes in FePS$_3$ within the $ab$ plane have similar energy while those related to the perpendicular direction are heavily affected.

## 3.2 Electronic Properties of FePS$_3$

### 3.2.1 LP and HP-I

We have also investigated the evolution of electronic properties along with the structural transition in response to external pressure. The projected density of states (PDOS) are calculated with DFT+$U$ (Fe $d$: $U = 2.5$ eV) methodology, using CASTEP code and then post-processed with the OptaDOS package [68, 69]. In the LP and HP-I phases, the spin-up and spin-down magnetic arrangements for all Fe atoms are equivalent as they both stabilize with long-range AFM order. Fig. 6 displays the PDOS involving the projection of Kohn-Sham states onto the spin-up and spin-down Fe 3$d$ orbitals, together with the ones for P 3$p$ and S 3$p$, for LP at 0.0 GPa and 4.0 GPa (b), and for HP-I at 4.0 GPa (c) and 10.0 GPa (d). The apparent perfect spin-up-down compensation is characteristic of net zero moments in AFM arrangements. It

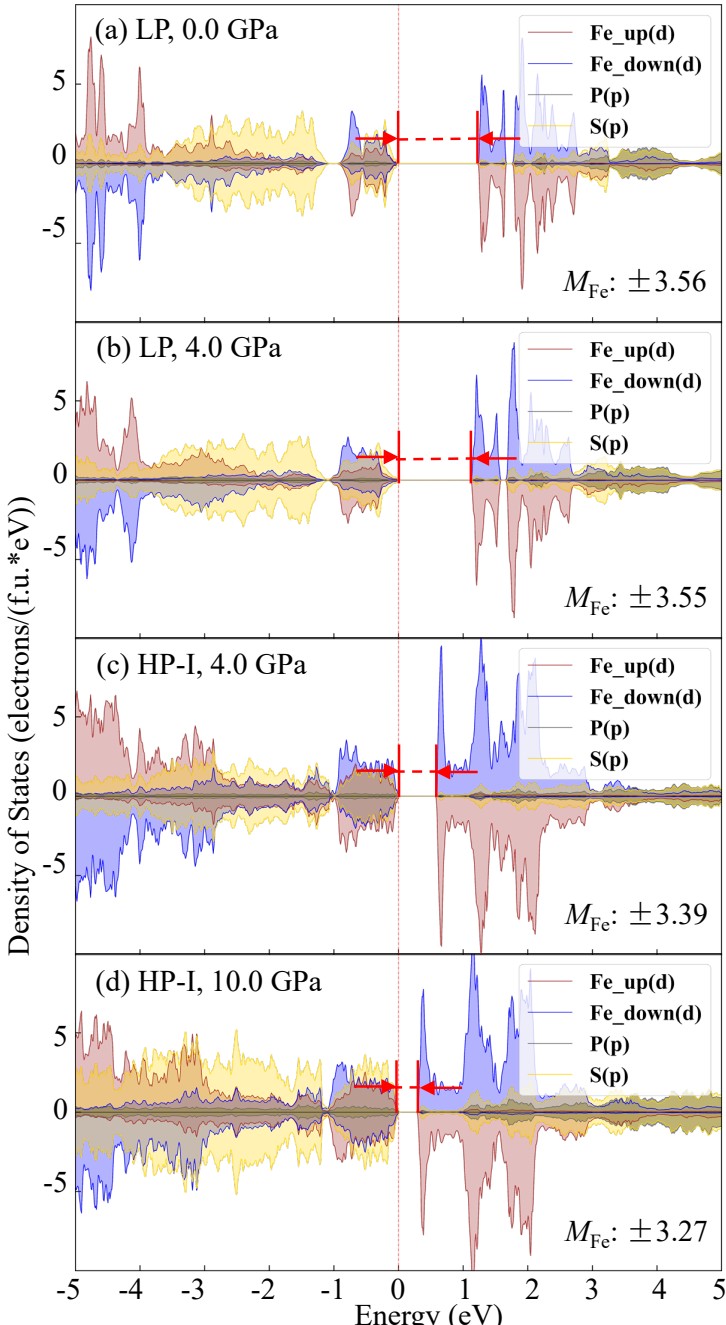

Figure 6: Projected density of states (PDOS) as calculated within DFT+$U$ (Fe $d$: $U = 2.5$ eV), post-processed with the OptaDOS package [68, 69], showing the projection of Kohn-Sham states onto the Fe 3$d$ states for spin-up (positive sign, red) and down (negative sign, blue). Grey PDOS indicate P 3$p$ and yellow is for S 3$p$. Panels (a) and (b) for LP at 0.0 GPa and 4.0 GPa (b), respectively, (c) and (d) show HP-I at 4.0 GPa and 10.0 GPa. $M_{\text{Fe}}$ refers to the final spin moments on Fe.

should be noted, however, that, if projected on separate Fe atoms, they show mostly up (or down) according to their net polarization obtained and displayed in Fig. 1.

The obtained electronic structure is consistent with the attributed Mott insulating state, with a clear split between occupied and unoccupied Fe $3d$ orbitals, the weight of the S anions staying clear of the bands around the Fermi level, thereby ruling out a charge-transfer character for the insulating state. This fact is especially clear for LP, but less clean cut for the HP-I phase, and in both cases, P states significantly mix with the Fe $3d$ states, complicating the modelling. This latter fact is probably what is behind the less idealized parameterizations needed in spin models [70], which give rise to the frustration effects that result in the observed spin arrangements, instead of the pure Néel state expected from the bipartite honeycomb Fe substructure. The obtained electronic structure for this phase is also consistent with the fact that the energy scales for the definition of spin moments, the high spin state for $Fe^{2+}$ ($S = 2$) and the effective spin-spin interactions dominate over any crystal field splitting parameter, $\Delta$, that would tend to reduce the net spin on individual Fe atoms.

The obtained band gap, as determined by the bottom of the conduction and the top of the valence Kohn-Sham bands, for the LP phase at 0.0 GPa is about 1.2 eV. This is qualitatively consistent with previous optical measurements of the band gap [71]. It is well known that band gaps are not quantitatively predicted from this level of theory. Although the DFT+$U$ correction (including sensible, phenomenological values of $U$) somewhat improves their reliability, we will follow them here qualitatively as support for consistency and an indicator of trends.

With increasing pressure, the band gap for LP gradually shrinks down to $\sim 1.1$ eV at 4.0 GPa. However, when LP transforms into HP-I, the band gap becomes significantly smaller, $\sim 0.56$ eV for the HP-I at 4.0 GPa. The HP-I phase remains insulating at the applied pressure $P = 10.0$ GPa with a reduced band gap of $\sim 0.27$ eV. Though the increased pressure would increase the crystal field splitting energy $\Delta$ and thus affect the competition between $\Delta$ and the exchange energy $J$, the high-spin state of $Fe^{2+}$ ($S = 2$) is still maintained in both the LP and HP-I phases within the pressure range being discussed.

### 3.2.2 HP-II and HP-II-$\beta$

Figure 7 shows the same PDOS decomposition as Fig. 6 now for $P = 11.0$ GPa and the HP-II phases. Panel (a) shows the HP-II-$\beta$ phase, which appears clearly beyond the insulator to metal transition, using the same level of theory as employed so far. No spin decomposition is obtained. The dimensionality difference, introduced by the distinctive interlayer P bonding behaviour, affects the electronic structure across the Fermi level. With continuous contributions from P and S states, the metallicity also significantly affecting the distribution of the $3d$ orbitals of Fe.

There is uncertainty in the most suitable choice for the value of the Hubbard $U$. The experimental IMT observed at higher pressures (see Fig. 1) seems to indicate stronger correlation effects than for other Fe compounds. In Fig. 7 we explore the effect of on-site Coulomb repulsion by tuning the Hubbard $U$ on Fe $d$ orbitals from 2.5 to 3.75 and 5.0 eV. The geometries are relaxed at 11.0 GPa with different $U$, and the corresponding PDOS are then calculated as shown. The increased $U$ values push the insulator to metal transition to higher pressures, in closer agreement with experiments, the insulating phase always remaining HP-I (panel c) with the same spin arrangements (inset). The increased electron-electron repulsion seems to inhibit the formation of strong chemical inter-layer P-P bonds, while retaining the long-range order (LRO) zigzag structures within planes. Interestingly, whenever the gap closes, we obtain the $\beta$ phase.

The fact that the calculations lose spin polarization upon metalization has to be properly interpreted. The DFT+$U$ calculations rely on the mean-field treatments of the electronic problem (even at the Hubbard level) and cannot describe short-range spin correlations. Coak

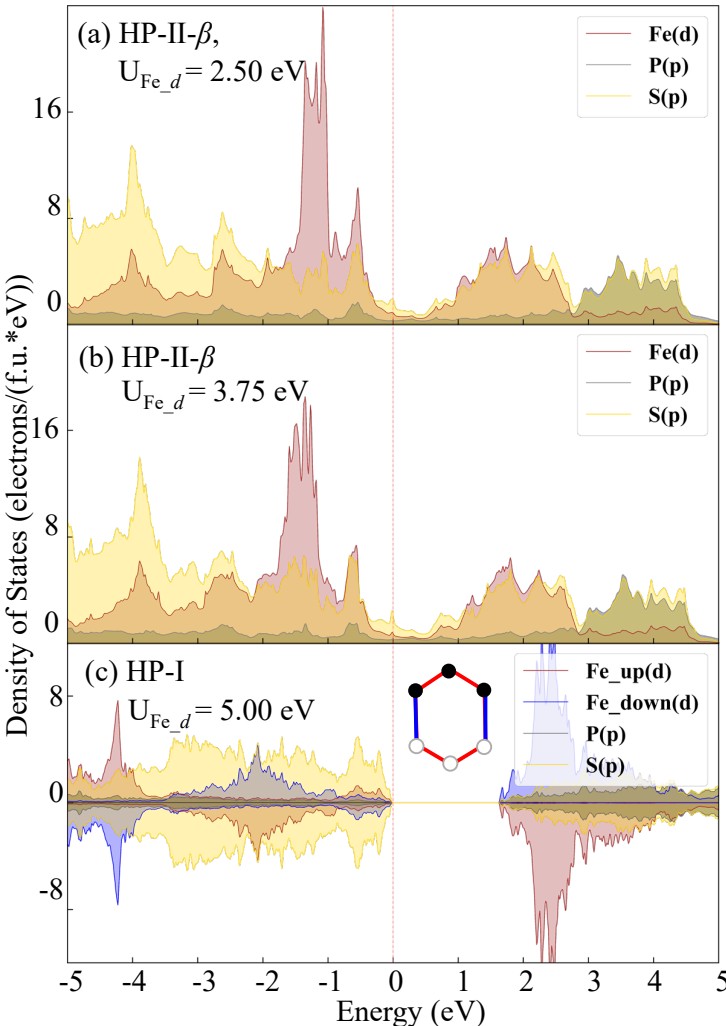

Figure 7: PDOS for HP-II-$\beta$ [panels (a) and (b)] and HP-I (c), respectively at $P = 11.0$ GPa (same conventions as for Fig. 6). Hubbard $U = 2.50$ eV (a), 3.75 eV (b), and 5.00 eV (c) on the Fe 3$d$ orbitals.

*et al.* [41] observe from their experiments that short-range order (SRO) magnetism persists in the high-pressure region, resembling that of the HP-I magnetic configuration. It is an interesting finding, compatible with the physics close to the Mott transition. There could be ways to query the calculations in that direction, by introducing constrained spins by hand at selected locations, and seeing the spin texture decaying from those sites in large-simulation-box settings. In an AFM situation the procedure is however quite arbitrary, since the individual magnetic moments develop for the electrons associated with a given atom, which is always an ill-posed proposition. Higher levels of theory, such as dynamical-mean-field theory (DMFT) [72] would also help in this regard, but they represent a substantially increased computational and theoretical effort, well beyond the scope of this work.

In addition to the displayed PDOS for various phases, Figs. 8 and 9 show the corresponding band-structure dispersion relations along paths crossing selected high-symmetry points. Fig. 8 shows these for LP and HP-I, from 0.0 GPa to 10.0 GPa. The $\Gamma$-A and M-L Brillouin-zone segments are along the $k_z$ direction, (nearly) perpendicular to the layer planes in real space. The bands along these directions are quite flat, as expected for weakly interacting 2D layers.

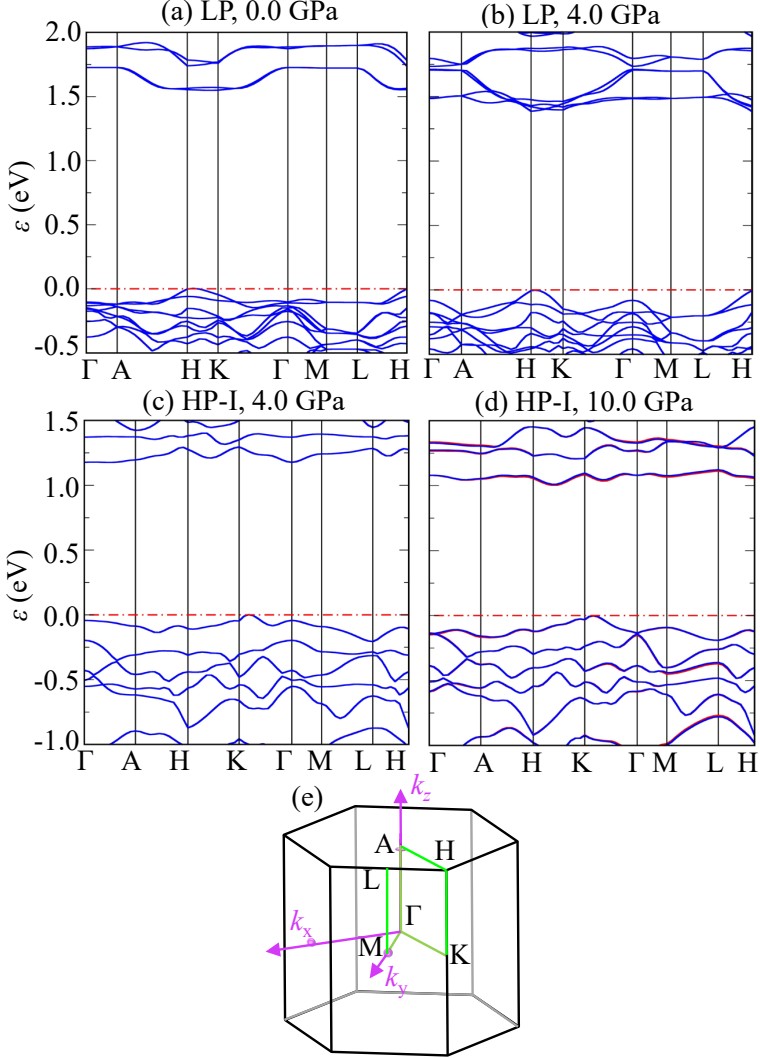

Figure 8: Band structures for (a) LP at 0 GPa and (b) 4.0 GPa, and for (c) HP-I at 4.0 GPa and (d) 10.0 GPa, and (e) a Brillouin zone diagram indicating the symmetry points defining the band dispersion plots.

Fig. 9 shows the band dispersion for the metallic high-pressure phases, very close to the insulator-to-metal transition ($P = 7.4$ GPa, as obtained for $U + 2.5$ eV), in both its originally proposed layered form HP-II and its 3D HP-II-$\beta$ version. The metallization is much clearer in the latter, the former representing a weak semimetal. In Fig. 9 (a), electron and hole pockets emerge near the Fermi level along the high symmetry path H-A. It suggests that the HP-II phase would transit through a semimetal phase before becoming fully metallic at slightly higher pressures. From our results, however, it is difficult to ascertain whether there could be such an intermediate region of stability.

Several bands cross the Fermi level in HP-II-$\beta$, as can be seen in Fig. 9 (b). Our results suggest that the metallicity would emerge at the high-pressure region regardless of the dimensionality collapse in the $c$ axis. Our calculations offer another scenario in this case. It should be noted that for the high-pressure phase reported in [31, 40] the band structure is actually very similar to the one we obtain for HP-II-$\beta$, in spite of their not reporting on the dimensionality change implied by the $\beta$ phase. The $c$-axis collapse and P-P interlayer bonds appearing in the HP-II-$\beta$ phase should affect electronic transport significantly. Further investigation is needed to understand the effect of dimensionality crossover on electronic transport. It would

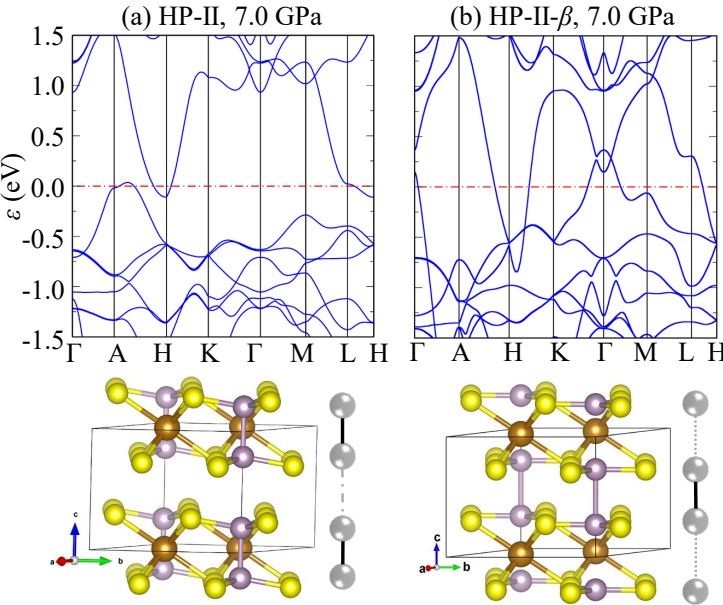

Figure 9: Band structures for (a) HP-II and (b) HP-II-$\beta$ phases at 7.0 GPa (both $P\bar{3}1m$), close to the transition point, with structural sketches under each corresponding panel.

require the deployment of more sophisticated techniques (like DMFT [72], whenever accessible to these system sizes) to describe the effect of the strong electron correlations induced by the strength of on-site Coulomb repulsion for $3d$ orbitals.

## 4 Conclusions

With AIRSS random structure search, we have reproduced the previously proposed LP, HP-I and HP-II phases, and a novel HP-II-$\beta$ phase. LP and HP-I crystallise in the monoclinic space group $C2/m$ while HP-II and HP-II-$\beta$ stabilize in the trigonal one $P\bar{3}1m$. The HP-II-$\beta$ is intrinsically different from HP-II in that the P atoms form stronger chemical bonds in-between the neighbouring layers. The full simulation of the enthalpy-pressure phase diagram, taking into consideration spin polarisation, is found to be consistent with the previous experimental observations.

We rationalize the coexistence region of the LP and HP-I phases by quantifying the energy barriers for neighbouring planes to slide against each other, regardless of the interplanar coupling being antiferromagnetic (LP) or ferromagnetic (HP-I). Despite the gradual decrease of the band gap size, both phases remain insulating in this pressure region.

At higher pressure, the symmetry crossover from $C2/m$ to $P\bar{3}1m$ takes place concurrent with the emergence of metallicity. The intra- or inter-layer P-P bonding affects the energy bands near the Fermi level, and might be responsible for a possibly different explanation for the origin of metallicity. The predicted dynamically stable HP-II-$\beta$ phase defines a scenario for FePS$_3$ turning metallic while becoming 3D-connected under high pressure.

The magnetic moments on the transition metal sites are suppressed when the system turns from quasi-2D to the 3D limit. The strength of the on-site Coulomb repulsion also influences the dimensionality change. The actual correlation strength deserves further investigation as it could facilitate quantitative theory in the low-dimensional $TMPX_3$ and other AFM Mott systems.

Considering the complexity of correlation effects in the system and the competition among exchange and anisotropy, more effort is needed to further our understanding of the nature of the insulator-to-metal transition and how dimensionality is involved in the process. This work has been carried out with the DFT+U technique, which is insensitive to environmental factors like nature of pressure medium. The theoretical modelling can be expected to guide future experimental explorations in the $TXP X_3$ compound family and could possibly be extended to other 2D magnets, where pressure or other tuning parameters could tune the phases to be novel metal or unconventional superconductors.

# Acknowledgement

The authors would like to thank C.J. Pickard, M.J. Coak, D.M. Jarvis, C.R.S. Haines, H. Hamidov, C. Liu, X. Zhang and A.R. Wildes for their generous help and discussions.

**Funding information** This work was carried out with the support of the Cambridge Service for Data-Driven Discovery (CSD3) and the UK Materials and Molecular Modelling Hub (MMM Hub). S.S.S. and S.D. acknowledge support from Department for Business, Energy and Industrial Strategy, UK (BEIS Grant Number: G115693), Department of Science, Technology and Innovation (DSTI Grant Number: G117669) and Cavendish Laboratory. S.D. acknowledges a scholarship to pursue doctoral research from the Cambridge Trust China Scholarship Council. S.C. acknowledges financial support from the Cambridge Trust and from the Winton Programme for the Physics of Sustainability. B.M. acknowledges financial support from a UKRI Future Leaders Fellowship (Grant No. MR/V023926/1), from the Gianna Angelopoulos Programme for Science, Technology, and Innovation, and from the Winton Programme for the Physics of Sustainability. E. A. acknowledges funding from Spanish MICIN through grant PID2019-107338RB- C61/AEI/10.13039/501100011033, as well as a María de Maeztu award to Nanogune, Grant CEX2020-001038-M funded by MCIN/AEI/ 10.13039/501100011033.

# A  Convergence tests

We have performed the convergence tests for the plane-wave cutoff energy ($E_c^{PW}$) and $\boldsymbol{k}$-points sampling within the framework of static single-point calculations using CASTEP. The LP phase of FePS$_3$ at ambient pressure has been chosen to perform the tests. The total energy dispersions as a function of $E_c^{PW}$ and the maximum spacing between each $\boldsymbol{k}$-point are displayed in Fig. 10. Based on the convergence tests, the $E_c^{PW} = 550$ eV and k-points sampling of 0.03 Å$^{-1}$ along each axis has been set for the rest of all calculations, including the high-pressure phases.

# B  Sensitivity of dimensional crossover at high pressure for various approximations

We also investigated the sensitivity of pressure-induced P-P from adjacent layers forming stronger bonds with respect to the choice of simulation parameters in detail. Considering the fact that FePS$_3$ is a layered vdW compound with Fe being able to host finite magnetic moments, we postulate a few starting scenarios in HP-II phase and then fully relax the structure with CASTEP. The P-P interatomic distances within each double sulfur layer and in between the neighbouring layers are evaluated from the post-optimized phases. Fig. 11 summarizes

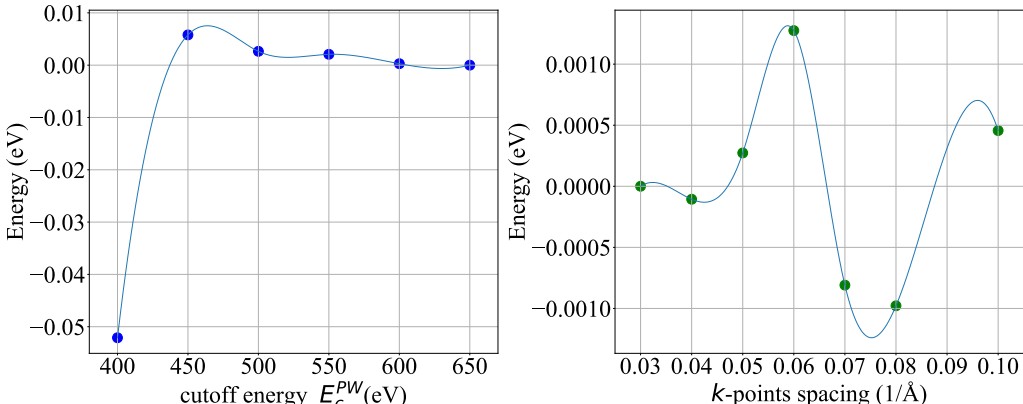

Figure 10: Convergence with plane-wave energy cutoff $E_c^{PW}$ of the total energy of the LP phase of $FePS_3$ at ambient pressure (left), and with $\boldsymbol{k}$-point spacing (right). The vertical axis of energy has been rescaled so that the energies calculated with the highest energy cutoff ($E_c^{PW} = 650$ eV) and the highest $\boldsymbol{k}$-point density (spacing = 0.03 1/Å ) corresponds to 0 eV.

the two P-P interatomic distances as a function of pressure, with the former in blue and the latter in green.

In Fig. 11 (a), the non-spin-polarized HP-II phase has been relaxed from 1 to 15 GPa with a step size of 2 GPa. The functional is chosen as PBE [47], and the vdW correction methodology follows the Tkatchenko-Scheffler (TS)'s approach [52]. There are the parameters that we have discussed and utilized for most of the calculations in the main text. We also compared the S. Grimme's semiempirical approach (2006) [73] (equivalent to the DFT-D2 in the Vienna *ab initio* simulation package (VASP) [74]) to tackle the vdW interactions. It can be seem from Fig. 11 (b) that the P-P interlayer dimerization with pressure will not be affected by the choice of vdW correction methodology qualitatively. Similarly, the PBESol [75] as functional has been explored in Fig. 11 (c), without qualitative effect on the transition.

In Fig. 11 (d-f), we explored the effect of Hubbard $U$ on Fe $d$ orbitals at 2.5, 3.75 and 5.0 eV. The strength of Hubbard $U$ has the most prominent influence. The transition point having been postponed to as high as around 17 GPa when $U$ is 5.0 eV. In addition, we also explored the effect of symmetry in Fig. 11 (g-i). We create the supercell with 4 f.u. inside one simulation cell from HP-II phase to allow for zigzag FM chains along $a$ axis being antiferromagnetically coupled to each other within each layer. The relaxed structures by symmetry analysis shall be classified as HP-I phases instead. The formation of interlayer P-P dimer is more sensitive to the correlation effect when the $C3$ rotation symmetry has been broken. The neighbouring layers remain to be gapped via vdW interaction.

To conclude, P atoms tend to form shorter and stronger bonds across neighbouring layers at higher pressure. The high-pressure phase stabilizes in HP-II-$\beta$ energetically within the DFT+U methodology. The choice of different vdW corrections to the final energy and density functionals barely affects the system to stabilize in the HP-II-$\beta$ at high-pressure region. Meanwhile, the Hubbard $U$ on Fe $d$ orbitals influences the formation of interlayer P-P bonds substantially, which is consistent with the picture of correlated electrons in $FePS_3$.



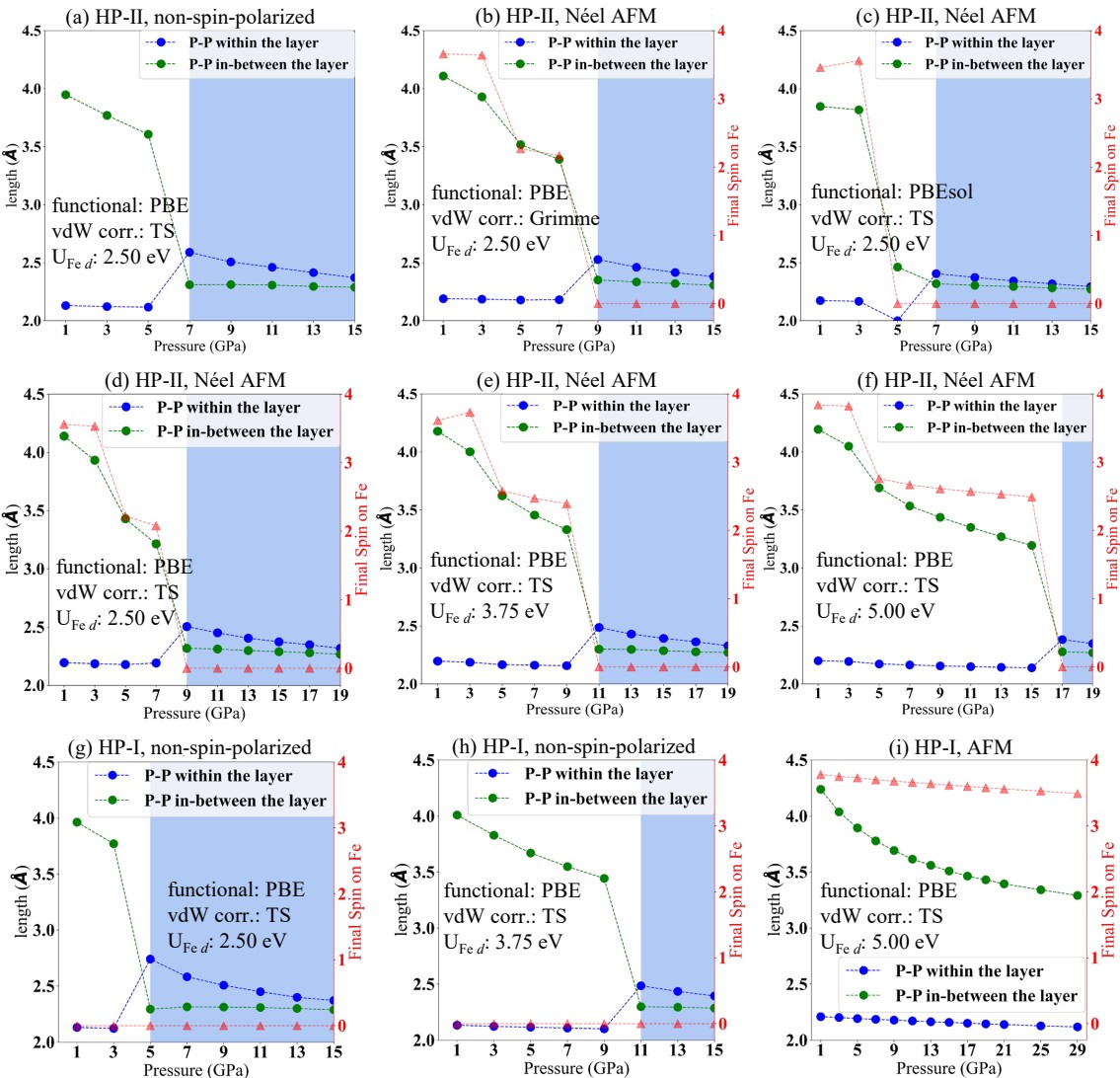

Figure 11: Evolution of P-P interatomic distances within (blue) and in-between (green) the layers in response to simulated hydrostatic pressure for the HP-II (a-f) and HP-I (g-i) phase under different simulation scenarios.

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
