# Peer review of "Pressure-induced transitions in FePS$_3$: Structural, magnetic and electronic properties"

_SciPost Physics, doi:SciPost Phys. 15, 020 (2023)_

## Round 2 · Referee Report · Anonymous (Referee 1) · 2023-3-6

Strengths

1-Topic of broad interest
2-Very detailed analysis
3-Interesting results

Weaknesses

1-Perhaps a little too long and redundant in some places

Report

The authors investigate the structural phases and electronic properties of FePS3 compound by means of density functional theory calculations. They perform a very detailed analysis of the system under pressure via a random structure search method to check what was found by experimental investigations and in order to predict possible new phases. With their investigation, the authors reproduce the known phases LP, HP-I and HP-II and they predict a new phase, namely HP-II-β . At ambient pressure the system is insulating, at pressure of around 14 GPa there is an inulator to metal transition, when the system changes structure from C2/m to P3 ̅1m in the HP-II phase. The new phase HP-II-β is predicted by the authors to be stable by means of phonon dispersion calculations. They also investigate the electronic properties and they present the evolution of the projected density of states as a function of the pressure and they show how the band gap becomes smaller at high pressures. The intra- or inter-layer P-P bonding affects the energy bands near the Fermi level, and might be responsible for a possibly different explanation for the origin of metallicity. The predicted dynamically stable HP-II-β phase defines a scenario for FePS3 turning metallic while becoming 3D-connected under high pressure. The magnetic moments on the transition metal atoms get suppressed when the system goes from the two-dimensional to three-dimensional limit.

I think the paper is well written and the results are interesting, I believe that it meets the acceptance criteria for publication as a SciPost article. The investigation of magnetism and structural properties of MPX3 compounds is a modern topic of great interest for applications and because these systems present metal to insulator transition and superconductivity phase too. These materials become fertile ground for exploring novel phases and emergent phenomena. Therefore I suggest the publication of this paper.

---

## Round 2 · Referee Report · Anonymous (Referee 2) · 2023-3-9

Strengths

Well-written work
Accurate ab-initio-based analysis
Potentially interesting results

Weaknesses

Introduction to be expanded
Number of figures elavated
Absence of theoretical modeling

Report

By means of a first-principles study that uses van der Waals and Hubbard U corrected density functional theory including a random structure search the Authors try to understand how the external pressure tunes the dimensionality, structural, electronic and magnetic properties in FePS3. This numerical computational study attempts to interpret the experimental coexistence of the low- and intermediate-pressure phases and predicts a novel high-pressure phase with distinctive dimensionality and different possible origins of metallicity. A detailed study is also performed looking at the modifications induced by the interlayer sliding and dimensionality. The paper is interesting, well written and contains new results that deserve to be published. Nevertheless, before publication the following comments and remarks have to be properly addressed:
1. The references in the Introduction Section are thin on the ground and should be tailored to the subject at hand. Thus, I suggest enlarging this section mentioning recent papers.
See for instance for TMPX3 compounds:
Phys. Rev. Research 4, 023256 (2022); Phys. Rev. B 106, 035137 (2022); J. Phys. Chem. C 126, 6791 (2022); 2D Mater. 10, 014008 (2023); Phys. Rev. B 107, 075423 (2023).
See for instance for pressure-induced superconductors:
Nat. Commun. 5, 5508 (2014); Nat. Commun. 12, 5436 (2021); Proc. Natl. Acad. Sci.118, e2108938118 (2021); npj Quantum Mater. 7, 93 (2022); Nature 615, 244 (2023)
2. The explanation or some comments need to clarify the nature of the insulator-to-metal transition and how dimensionality is involved in the material under study.
3. Since the theoretical approach relies on DFT+U technique, I suggest to try an effort to theoretically model TXPX3, as done in some of the above mentioned papers.

Requested changes

Introduction to be expanded
Theoretical model to be proposed

---

## Round 3 · Author Response

Claudio Attaccalite on 2023-04-06 [id 3561]
I publish the authors' communication in this comment, so that the referees can read it and formulate a possible response.
/===================================/
Dear Dr. Claudio Attaccalite,
Thank you for your insightful review of our manuscript and the opportunity to revise our paper. We appreciate the thorough assessment and constructive feedback provided by both referees. We have carefully considered their comments and suggestions and made necessary changes in the revised manuscript.
Regarding Referee 1's report, we have addressed the concerns raised, such as removing redundancy in the text and improving clarity. We appreciate the referee’s positive comments regarding the topic and results presented in our work, and we thank them for suggesting our work for publication.
In response to Referee 2's report,
1 ) We have expanded the introduction section to include recent papers on TMPX3 compounds and pressure-induced superconductors, as suggested by the second referee’s report: We have added [2D Mater. 10, 014008 (2023)] as an example of “clean energy generation related with photocatalytic water splitting applications”. We have added [J. Phys. Chem. C 126, 6791 (2022); Phys. Rev. B 106, 035137 (2022)] as monolayer manufacturing references. We have added [Nat. Commun. 5, 5508 (2014); Nat. Commun. 12, 5436 (2021); Proc. Natl. Acad. Sci.118, e2108938118 (2021); npj Quantum Mater. 7, 93 (2022)] to expand the pressure-induced superconductivity near antiferromagnetic order. We did not add [Nature 615, 244 (2023)] as the debate regarding this work is still underway. We have added [Phys. Rev. B 107, 075423 (2023)] to address the complexity of magnetic interactions in TMPX3.
2) In our paper, we have illustrated how the band gap shrinks with increasing pressure in LP and HP-I phases and the insulator-to-metal transition happens when HP-II-beta phase becomes the most energetically favourable phase. The dimensionality is explored as a function of the inter- and intralayer P-P distances. The HP-II-beta phase is 3D-connected by contrast to the HP-II phase, of which the interlayer distance gets suppressed but the van der Waals gap remains.
3) While we recognize the importance of establishing a theoretical model for FePS3, this is beyond the scope of the current study. We cannot delineate a simple model which addresses the insulator-to-metal transition, the magnetic ordering evolution and P-P interlayering bonding at the same time within the DFT+U framework. The models proposed by some of the previous studies are focusing on the magnetic interactions at ambient pressure. However, we will keep this suggestion in mind for the future studies.
Overall, we believe that these revisions have strengthened the manuscript and addressed the referees’ concerns. We hope that these revisions adequately address your concerns and that you find our manuscript suitable for publication in SciPost Physics.
Thank you again for considering our manuscript for publication. We look forward to hearing your decision.
Yours sincerely, Shiyu Deng, Siyu Chen, Bartomeu Monserrat, Emilio Artacho and Siddharth Saxena
Author: Shiyu Deng on 2023-04-11 [id 3568]
(in reply to Claudio Attaccalite on 2023-04-06 [id 3561])Dear Dr. Claudio Attaccalite,
Thank you for sharing the positive feedback from the referees. We are glad to hear that they found our revisions satisfactory and that the manuscript has been improved.
Regarding the suggestion to insert the author names of ref. 20 and cite [J. Phys. Chem. C 126, 6791 (2022)], we apologize for the oversight and have made the necessary changes in the revised manuscript. The updated version has been published via arXiv: 2209.05353v5 (https://arxiv.org/abs/2209.05353) and is ready for resubmission.
We appreciate your consideration of our manuscript for publication in SciPost Physics and look forward to hearing your final decision.
Yours sincerely,
Shiyu Deng, Siyu Chen, Bartomeu Monserrat, Emilio Artacho and Siddharth Saxena
Anonymous on 2023-04-06 [id 3562]
(in reply to Claudio Attaccalite on 2023-04-06 [id 3561])The authors have addressed all points raised by the Referees. Their answers are satisfying, the changes made to the manuscript appropriate, and then the revised version of the paper definitely improved. I only suggest suggest to insert the author names of ref. 20 and cite ref. J. Phys. Chem. C 126, 6791 (2022) mentioned in the reply but missed in the text.
Therefore, I recommend publication of the manuscript in SciPost Physics.

---

## Editorial Decision

published